# Photoproducts of the Photodynamic Therapy Agent Verteporfin Identified via Laser Interfaced Mass Spectrometry

**DOI:** 10.3390/molecules25225280

**Published:** 2020-11-12

**Authors:** Chris Furlan, Jacob A. Berenbeim, Caroline E. H. Dessent

**Affiliations:** Department of Chemistry, University of York, Heslington York YO10 5DD, UK; cf1059@york.ac.uk (C.F.); jacob.berenbeim@york.ac.uk (J.A.B.)

**Keywords:** verteporfin, photosensitizer, photo dynamic therapy, PDT, photofragments, photofragmentation channels, mass spectrometry, laser spectroscopy, photolysis

## Abstract

Verteporfin, a free base benzoporphyrin derivative monoacid ring A, is a photosensitizing drug for photodynamic therapy (PDT) used in the treatment of the wet form of macular degeneration and activated by red light of 689 nm. Here, we present the first direct study of its photofragmentation channels in the gas phase, conducted using a laser interfaced mass spectrometer across a broad photoexcitation range from 250 to 790 nm. The photofragmentation channels are compared with the collision-induced dissociation (CID) products revealing similar dissociation pathways characterized by the loss of the carboxyl and ester groups. Complementary solution-phase photolysis experiments indicate that photobleaching occurs in verteporfin in acetonitrile; a notable conclusion, as photoinduced activity in Verteporfin was not thought to occur in homogenous solvent conditions. These results provide unique new information on the thermal break-down products and photoproducts of this light-triggered drug.

## 1. Introduction

Verteporfin (Figure 1) is a porphyrin base used as a photosensitizer drug in the treatment of macular degeneration via photodynamic therapy (PDT) [1]. PDT is a treatment modality that has been used in the successful treatment of several diseases and disorders and uses a combination of a selectively localized light-sensitive drug (photosensitizer) and light of an appropriate wavelength, to produce toxic species that can interact with cellular constituents causing biochemical disruption to the cell [2]. The proposed PDT activation mechanisms are considered to be based on the reaction of the singlet or triplet light-activated producer with the substrate or solvent (Type I) or with molecular oxygen generating highly reactive singlet oxygen (^1^O_2_) and/or reactive oxygen species (ROS) (Type II) [3,4,5].

However, in photophysical processes, it is very common to produce photofragments derived by bond breakage and rearrangement of the light-sensitive molecule, and it is possible that the photofragmentation products themselves may react with the cellular biomolecules contributing to cytotoxicity. Currently, there are no data on the photofragmentation pathways of verteporfin and sparse information is available regarding the identity of its photoproducts. One of the main relevant studies was carried out by Gillies et al. [6], who reported the photophysical properties of a photoproduct generated in vitro from Verteporfin (benzoporphyrin derivative monoacid ring A, BPD-MA) in fetal calf serum (FCS) solutions under 694 nm light irradiation. They found that the changes in the absorption and fluorescence spectrum before and after irradiation are consistent with the production of a photoproduct that was proposed to be a hydroxyaldehyde and/or formyl derivative generated by cycloaddition of singlet oxygen and free radical mechanisms, respectively, similarly to the proposed photoproducts of protoporphyrin IX [7]. Furthermore, according to Gillies et al., no photoproduct formation occurred for BPD-MA in homogenous organic solutions suggesting that photoproduct formation occurred only when BPD-MA is bound to proteins in fetal calf solutions (FCS) with a high dependence on the presence of oxygen.

An additional study on Verteporfin (BPD-MA) and protoporphyrin IX (PpIX) in vivo using fluorescence spectroscopy by Iinuma et al. [8] demonstrated that in vivo, photobleaching rates and ^1^O_2_ generation are not directly related suggesting that it is possible that two different photobleaching processes exist for both PpIX and BPD-MA: One being oxygen-dependent, and the other oxygen-independent, so that the measured photobleaching decay constant is a composite of the two processes. No photofragments or the photodegradation pathways were identified, however, although the authors did underline the complexity of the photobleaching process. Another fluorescence spectroscopy study on a Verteporfin-like photosensitizer by Harvey Lui et al. [9] showed that a stable and quantifiable PDT photoproduct that fluoresces at ~650 nm can be reliably measured in vivo from mouse skin, and used this to calculate the photoproduct score used in PDT dosimetry. However, again in this study, no more details on the nature of the photoproduct or the mechanism of its production were provided. We note that Berezin and coworkers recently studied the photostability of a group of related porphyrinoids, finding that their photostability depended on the degree of non-planarity of the macrocycle, as well as the symmetry of the protonated form present in proton-donating solvents [10].

In the current work, we use a novel approach to measure the direct photoproducts of Verteporfin. We perform measurements away from the complications of the solution-phase environment, by applying laser interfaced mass spectrometry (LIMS) to verteporfin as an isolated gas phase ion [11,12,13]. In this approach, the molecular system of interest is introduced to the gas phase from solution by electrospray ionization, mass-selected, and then isolated in an ion trap where it can interact with a laser beam. Any resulting photoproducts are then directly associated with the photodecay of the isolated molecule and can be identified using mass spectrometry. This general approach to using gas phase measurements to probe the intrinsic photodegradation pathways of photopharmaceutical molecules has been applied previously in our group to better understand the photochemistry of systems including photoCORMs and non-natural nucleobases [14,15,16,17]. In the current work, we also present preliminary results obtained by photolyzing a MeCN solution of Verteporfin, using mass spectrometric analysis to identify photoproducts.

The gas-phase photochemistry of several porphyrin systems have been studied previously by Brøndsted Nielsen and co-workers to obtain photodissociation and gas phase absorption spectra that are independent of the varying perturbations that can be encountered in a solution phase environment [18]. Action spectra for chlorophyll pigments revealed their intrinsic electronic properties [19,20,21,22], while photodissociation mass spectrum of Fe(III)-heme cations allowed identification of the photoproducts produced in the gas phase [18]. Kappes and co-workers have performed similar photodissociation spectroscopy measurements and photoelectron spectroscopy on gaseous meso-tetra(4-sulfonatophenyl)porphyrin (TPPS) multianions to probe their photo-decay channels [23,24]. Very recently, Weber and co-workers have studied the electronic spectra of cryogenically prepared protoporphyrin IX ions in vacuo, observing deprotonation-induced Stark shifts [25]. We note that collision-induced dissociation (CID) mass spectrometry has also been applied to study the gas phase thermal dissociation pathways of tetraphenyl iron and manganese porphyrin ions [26].

## 2. Results

### 2.1. Collision Induced Dissociation

Prior to commencing the photoexcitation measurements, collision-induced dissociation (CID) mass spectrometry was used to probe the thermal breakdown pathways of the molecule (Low-energy CID in a quadrupole ion-trap occurs via multiple low-energy collisions, resulting in the molecular ion breaking up along the same pathways as would occur upon heating [27]). Electrospray ionization of solutions of verteporfin resulted in strong production of the protonated molecule (*m*/*z* 719.3), which was isolated in the quadrupole ion trap and subjected to CID. We expect that the protonation site will be an N-atom of the pyrrolenine type sub-rings not already bound to hydrogen [28]. At around a collision offset voltage of 0.7 V, all the observed CID fragments are present (Appendix A), and their relative abundances do not change upon increasing the collision voltage further. Protonated verteporfin (precursor ion, *m*/*z* 719.3) is observed to fragment by losing masses up to 132 u generating six positively-charged verteporfin derivates. Analysis of the mass differences indicate that protonated verteporfin fragments through loss of neutral, even electron species derived from the loss of its (side chain) functional groups. The observation that CID of protonated verteporfin leads to the loss of a variety of functional groups is consistent with the molecule being protonated at the N-atom position, rather than on a single position on one of the side chains.

Table 1 lists the neutral fragments lost from protonated verteporfin upon CID, with the associated proposed fragmentation mechanisms given in Figure 2 (It is important to note that other mechanisms could lead to similar losses; the ones presented here were chosen based on chemical considerations (i.e., expected stability of the final products). The major CID product arises from a loss of methyl acetate (63%) followed by monomethyl succinate (14%), acetic acid (13%), methanol (4.5%), methyl propanoate (3.5%), and dimethyl oxalate (1.7%). The monomethyl succinate and dimethyl oxalate may derive from a combination of two radical fragments generated in proximity as shown in Figure 2b,f. Recombination of these radical fragments is thermodynamically favorable, and was hypothesized to occur in recombination of phenyl radicals in the CID dissociation of tetraphenyl iron and manganese porphyrins [26]. However, non-concerted loss of two radicals cannot be excluded in our experiment as the neutral fragments are not directly detected.

### 2.2. Gas Phase Photofragments

Protonated verteporfin was subjected to laser excitation in the ion trap of the laser interfaced mass spectrometer, and the resulting photofragments analyzed. In these experiments, mass spectra are automatically acquired at each scan step through the laser excitation range of 250 to 790 nm. The maximum photofragmentation yield is observed to occur at 398 nm, corresponding to the gas phase Soret peak (see below). Photofragmentation yields fall off considerably at wavelengths longer than 500 nm (the q-band region: Appendix A), as multiple visible photons would be needed to induce bond breaking at these energies [22,29].

When protonated verteporfin is excited in the region of its gas-phase Soret band (300–470 nm), numerous photofragments are observed (Figure 3a), including all of the fragments observed via CID. The most intense photofragment observed corresponds to loss of methyl acetate (*m*/*z* 645.3), mirroring the CID results.

Table 2 lists the fragments observed at the two different photoexcitation energies (398 nm and 758 nm), along with those from the CID experiment for comparison. UV light leads to more efficient photofragmentation as demonstrated by significantly higher ion signal intensities compared to visible/IR photon excitation. Notably, many additional minor photofragments are observed upon photoexcitation that are not observed upon CID, e.g., *m*/*z* 613.2, 571.2, 513.2, and 499.2. The presence of these photofragments indicates a complex photophysical picture, and multiple dissociation pathways [30]. Comparing the Soret-band (398 nm) and Q-band (730–790 nm) photofragmentation mass spectra, photofragmentation into lighter mass fragments is more prevalent at the higher photon energy, consistent with secondary fragmentation when higher excess energy is present [11,13,15].

The gas phase absorption spectrum of protonated verteporfin measured between 250 and 600 nm via laser interfaced mass spectrometry is presented in Figure 4a. The spectrum is acquired via photodepletion of the mass-selected protonated Verteporfin ion, i.e., by measuring the reduction in intensity of the precursor ion as a function of laser wavelength. This can be considered equivalent to the gas phase absorption spectrum subject to a number of conditions as discussed in detail previously [13,15,30]. Indeed, the gas phase absorption spectrum displays the same band profile as the solution-phase absorption spectrum (Appendix A). It is notable that the gaseous verteporfin Soret band peaks at 390 nm whereas the equivalent solution phase feature appears at 413 nm (Appendix A). Such a gas-phase to solution phase red shift of the Soret band is typical of similar porphyrin systems [18,20].

While the Soret band peaks at 390 nm, a shoulder on the band appears to be present at 430 nm. This feature can be assigned to a vibronic progression, which is evident in higher-resolution spectra of porphyrins [25]. Notably, this shoulder feature is also evident in the photofragment production spectra (*m*/*z* 687.3, 659.3, 645.3), which are shown with the gaseous absorption spectrum in Figure 4. The photofragment production spectra of the *m*/*z* 687.3, 659.3, 645.3 fragments all have similar profiles, which closely resemble the photodepletion spectra, except for an additional small band observed at 270 nm which appears more prominently in the photofragment production spectra than the absorption spectrum (photodepletion spectra typically display lower signal to noise since they are recorded in depletion.) The similar profiles of the photofragment and absorption spectra confirm that the fragments are produced by direct photodecay of the excited state molecule.

### 2.3. Photolysis

The photofragmentation products of verteporfin in solution (acetonitrile) were studied using photolysis cells to gain some preliminary insight into how the photoproducts of the isolated gas phase molecule change in a solution phase environment. Photolysed solutions were analyzed using ESI-MS to identify the photoproducts. Figure 5 presents the ESI-MS spectra obtained from the solutions exposed to light at 310, 365, and 689 nm. The first noticeable feature of these spectra is the presence of only two photofragments of significant intensity in addition to the protonated verteporfin mass peak (*m*/*z* 719.3). These peaks, whose identity is discussed further below, are absent in the mass spectra of both the pure solvent and non-photolyzed verteporfin control solutions. The low number of fragments contrasts with the much richer gas-phase fragmentation mass spectra (Figure 3). There are a number of explanations of why fewer photofragments are observed for solution photolysis versus gas phase photolysis. The gas phase photodissociation experiment takes place in a “zero background” environment, which allows efficient measurement of very low-intensity photofragments. Ionization efficiency of photofragments formed in solution could vary considerably. In addition, in solution low-intensity photofragments may react with the solvent, oxygen or other photoproducts, and hence branch into even less intense photoproducts. Furthermore, solution phase reactions that occur after photoexcitation of the chromophore can exist in equilibrium and lead to the most thermodynamically stable products. It should be recognized that a primary photoproduct can be photoexcited, and photofragment into a secondary photoproduct. This process cannot occur in the gas phase as trapped ions are only subject to single-photon excitation in our experiment [13,30].

Prolonged photolysis of the Verteporfin solution leads to complete photobleaching (lack of absorption of the solution in the UV–Vis), indicating that the aromaticity of the verteporfin macrocycle is largely destroyed upon photolysis. Photobleaching after extended irradiation (t > 4 h) was evident both in the lack of absorption in the UV–Vis when a solution-phase spectrum was subsequently recorded, and upon visual inspection of the photolysed solution which discolored from pale yellow to transparent (Appendix A). This result is in line with previous results for verteporfin [31].

At both 310 and 365 nm, the major fragmentation product observed (*m*/*z* 535.2) is probably derived from the loss of the functional groups similar to the CID and reaction with oxygen with loss of the highly conjugate structure to account for the lack of UV–Vis absorption [32]. The only other significant photofragment observed was the *m*/*z* 413.3 fragment, which was relatively more abundant upon 365 nm photolysis compared to 310 nm. Two photofragments were observed following excitation at 698 nm after 20 min exposure. Intriguingly, the main fragment at this excitation energy appeared at *m*/*z* 551.2, therefore corresponded to a photoproduct that was not identified upon UV photolysis. Tentative structures that can be assigned to the solution phase photoproducts are shown in Figure 6; the photofragments at *m*/*z* 535.2 and 551.2 are proposed to be derived from the loss of multiple fragments, which were lost individually in the gas phase experiment, followed by reaction with oxygen.

## 3. Discussion

The CID experiment shows that the thermal fragments arise from the loss of neutral methyl acetate (63%) followed by monomethyl succinate (14%), acetic acid (13%), methanol (4.5%), methyl propanoate (3.5%), and dimethyl oxalate (1.7%). These results suggest that the functional groups attached to the macrocycle are easily lost and an intermediate radical verteporfin moiety may be involved in some fragmentation pathways (Figure 2). These same fragmentation products are observed also in the gas phase photoexcitation experiment using the laser interfaced mass spectrometer, however in the latter, additional photoproducts were observed at lower *m*/*z* indicating photochemically distinct fragmentation pathways. We note that in both CID and gas phase photo-fragmentation experiments, there is a dominant dissociation mechanism that involves the loss of a methyl acetate molecule (*m*/*z* 645.3).

Calvo et al. studied the photophysical lifetimes of protonated protoporphyrin IX in vacuo in an ion storage ring [33]. They found that photoexcitation (390, 415, and 532 nm) lead to a triplet state lifetime of 6 ms and a quantum yield of 0.7, which is close to that of the free base and monocation in solution. The other decay channel observed, which experienced direct decay to the electronic ground state and subsequent dissociation of vibrationally excited ions, was much faster than triplet–singlet intersystem crossing. If analogous photophysics are present for the protonated verteporfin system, this would explain our observation of a mixture of photofragments that are identical to those observed upon CID (thermal decay is equivalent to dissociation of vibrationally excited ions), arising from direct singlet decay, and the “photochemical” photofragments which arise due to evolution of the triplet excited state.

ESI mass spectral analysis of photolysed solutions has been shown to be a useful tool for identifying photoproducts. For example, solution-phase photodegradation with ESI-MS identification of photoproducts was used in studies of quercetin, a food pigment, and fluopyran, a fungicide [34,35]. Similarly, ESI-MS analysis of the photolysed solutions of verteporfin in this study led to the identification of a number of photoproducts, tentatively assigned as products associated with the loss of macrocycle conjugation likely due to reaction with oxygen. These photoproducts are consistent with the observed discoloration of the solution (photobleaching). (Future experiments performed on solutions in the absence of oxygen will be useful to test this hypothesis.) The gas phase experiments suggest that the functional groups attached to the macrocycle can be easily lost after interaction with light, and the lower mass photoproducts identified from solution photolysis are consistent with multiple photon interactions leading to removal of multiple functional groups and rupture of the macrocycle.

This study demonstrated that the gas phase fragmentation mechanisms of benzoporphyrin derivatives are relatively easy to interpret, and therefore have potential to be used to guide the assignment of the complex photochemical processes that can occur in solution. That no intermediate functional losses were seen for *m*/*z* 535 may support kinetically driven reaction rates for verteporfin and suggest that transient spectroscopic studies may be needed to provide further information on the reaction mechanisms. Nonetheless, the gas-phase, in vacuo measurements show that photo-fragmentation processes can occur without the involvement oxygen and therefore the directly produced fragments (Figure 2) may react with the biomolecules and/or solvent. Overall, even if the proposed PDT activation mechanism of verteporfin, is considered to be mainly based on the generation of highly reactive singlet oxygen (^1^O_2_) through interactions of photosensitizer, light, and oxygen (^3^O_2_) [1], a small fraction of the activated photosensitizer appears to react with the surrounding solution components, hence potentially demonstrating a pathway for an enhanced cytotoxic effect.

## 4. Materials and Methods

### 4.1. Reagents and Electrospray

Verteporfin (≥94% HPLC grade) was purchased from Sigma Aldrich (St. Louis, MO, USA) and used without further purification. The solvent used was acetonitrile purchased from Sigma Aldrich (St. Louis, MO, USA). Solutions of verteporfin 10^−5^ M in MeCN were prepared and electrosprayed in the commercial Brucker AmaZon (Bruker Daltronics Inc., Billerica, MA, USA) quadrupole ion-trap mass spectrometer. These solutions and the verteporfin powder were stored at −20 °C in a dry environment. For the timescale of the gas phase experiments (≤2 h), it was verified to be stable at room temperature if protected from light, by observing that its mass spectrum remained unchanged.

### 4.2. CID

The first isotopic peak corresponding to the protonated verteporfin molecule was isolated (*m*/*z* 719.3) prior to CID, and the ion signal had good stability and intensity. CID in the quadrupole ion trap occurs via multiple low-energy collisions and is equivalent to thermal heating of the ion [27]. The AmaZon instrument was operated at 4 μL/min with the following parameters; ESI capillary −5000 V; End plate offset −700 V; ion transfer tube temperature, 120 °C; accumulation time 8 ms; fragmentation cut-off: *m*/*z* 200; fragmentation time: 100 ms. Note that fragments with masses <250 u will not be detected as they fall outside of the mass range of the mass spectrometer for these experiments.

### 4.3. Gas Phase Photofragmentation

Gas phase photodissociation experiments were conducted in an AmaZon SL dual funnel electrospray ionization quadrupole ion trap (ESI-QIT) mass spectrometer (Bruker Daltonics Inc., Billerica, MA, USA), which was modified to allow LIMS [15,36]. Photons were produced by an Nd:YAG (10 Hz, Surelite, Amplitude Laser Group, San Jose, CA, USA) pumped OPO (Horizon, Amplitude Laser Group, San Jose, CA, USA) laser. Solutions of verteporfin 10^−5^ M in MeCN were prepared and stored at −20 °C in a dry environment. These were electrosprayed using typical instrumental parameters (nebulizing gas pressure of 6.0 psi; injection rate of 0.30 mL/h; drying gas flow rate of 6.0 L/min), and run in positive ion mode at a capillary temperature of 120 °C. Photofragmentation experiments were conducted with an ion accumulation time of 10 ms and a fragmentation time of 100 ms, thereby ensuring that each mass-selected ion packet interacted with one laser pulse, minimizing the likelihood of multiphoton events. The first isotopic peak of the protonated verteporfin (*m*/*z* 719.3) was mass selected and stored in the ion trap where it was excited by UV–Vis laser pulse leading to fragmentation providing an action absorption spectrum by photodepletion (PD). Photodepletion and photofragmentation yields were determined as in previous studies [15,36].

### 4.4. Photolysis

The photolysis experiments were conducted with a home-built photolysis cells. All the photolysis experiments were conducted in solutions of verteporfin at 10^−3^ M in MeCN (not purged), and the samples were irradiated for 2 h. (All solution photolysis experiments were performed in acetonitrile, since this solvent was found to be the optimum solvent for electrospraying verteporfin.) After irradiation, the mass spectra of the solutions were taken to identify the fragmentation products. Both the photolysis cells have a 1.2 cm × 1.2 cm cavity with LEDS where the cuvette is inserted. One photolysis cell is provided with 4 LEDs peaking at 365 nm and provides an optical output power of 1200 mW at 700 mA. The second photolysis cell contains 8 LED’s, peaking at 310 nm providing an optical output power of 1.2 mW at 20 mA. A black plastic lid of thickness 1.1 cm was placed over the cuvettes in both UV sources to ensure that no light could escape.

A second experiment was performed using the OPO laser as the photon source; the wavelength chosen was 689 nm corresponding to the peak used in PTD, and the sample was irradiated for 20 min with a laser pulse of 0.12 mJ.

## 5. Conclusions

Laser photodissociation spectroscopy has been conducted on verteporfin as an isolated gas phase molecule for the first time to measure the direct photofragmentation products. It was found to photodegrade with loss of its carboxyl and ester side-chain functional groups. Solution-phase photolysis products were characterized using electrospray ionization mass spectrometry and assigned with the aid of the gas-phase measurements. The results presented illustrate how laser-interfaced mass spectrometry is a useful new tool for characterizing photoproducts and photodecay pathways, with potential for application to a range of photopharmaceutical molecules [37,38].

## Figures and Tables

**Figure 1 molecules-25-05280-f001:**
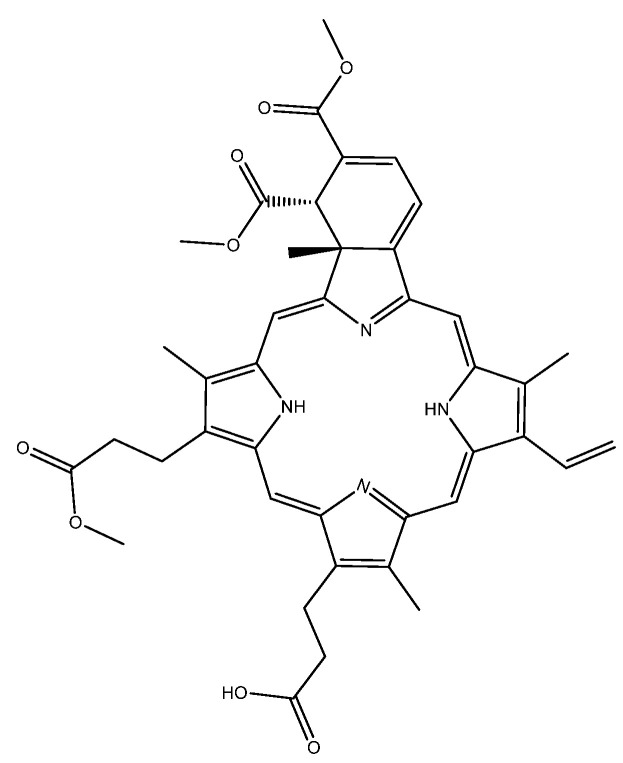
Schematic of a verteporfin molecule.

**Figure 2 molecules-25-05280-f002:**
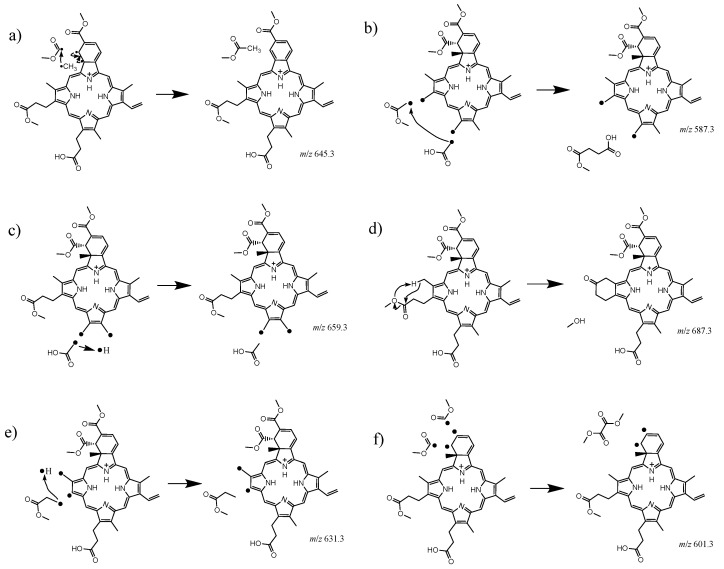
Proposed CID dissociation pathways associated with the peaks corresponding to (**a**) the major fragment observed at *m*/*z* 645.3 (63% abundancy), (**b**) the fragment at *m*/*z* 587.3 (14%), (**c**) the fragment at *m*/*z* 659.3 (13%), (**d**) the fragment at *m*/*z* 687.3 (4.5%), (**e**) the fragment at *m*/*z* 631.3 (3.5%), and (**f**) the fragment at *m*/*z* 601.3 (1.7%).

**Figure 3 molecules-25-05280-f003:**
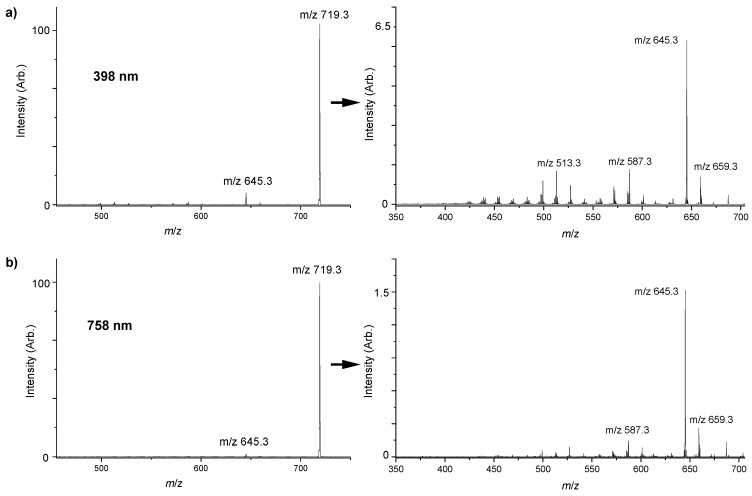
(**a**) Photofragment mass spectrum of protonated verteporfin, excited at photodepletion maxima at 398 nm. Laser pulse 0.05 mJ. (**b**) Photofragment mass spectrum of protonated verteporfin, excited at 758 nm. Laser pulse 0.20 mJ. The spectra on the right are an expansion of the respective spectrum on the left to allow the visualization of the minor fragment peaks.

**Figure 4 molecules-25-05280-f004:**
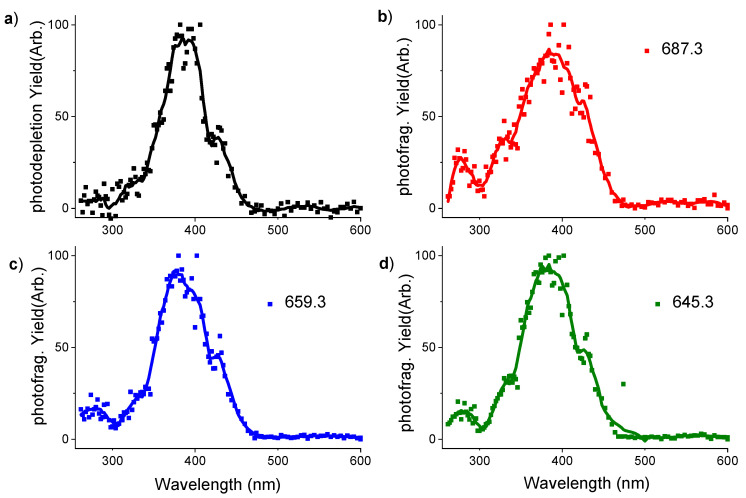
Photodepletion spectra of protonated verteporfin (**a**) and 3 photofragmentation spectra as examples *m*/*z* 687.3 (**b**), 659.3 (**c**), 645.3 (**d**). Laser pulse 0.05 mJ (400–260 nm), 0.3 mJ (402–600 nm), and *m*/*z* 200 cut-off.

**Figure 5 molecules-25-05280-f005:**
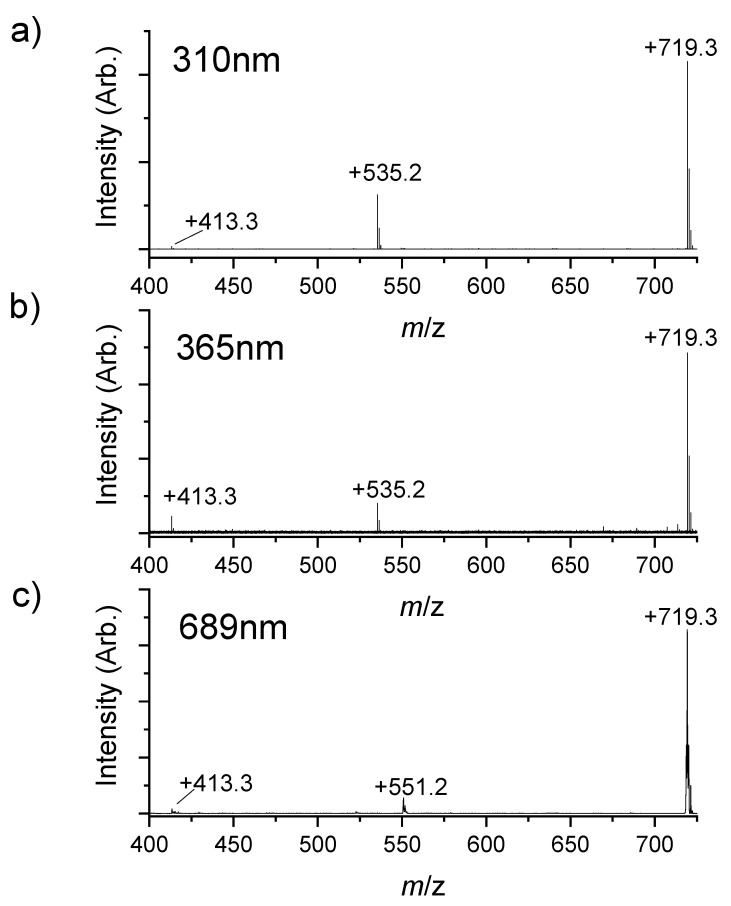
Mass spectra recorded after (**a**) 2 h exposure at 310 nm, (**b**) 2 h exposure at 365 nm, and (**c**) 20 min exposure at 689 nm.

**Figure 6 molecules-25-05280-f006:**
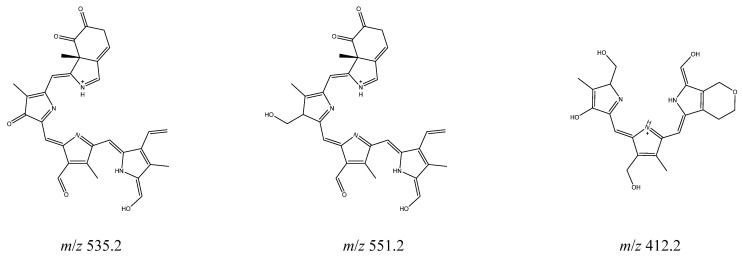
Proposed structures of the photofragments observed following solution photolysis. The fragment *m*/*z* 412.2 structure is proposed to attach a H atom to become the *m*/*z* 413.2 ion observed upon ESI. (The lowest energy position for this H atom to attach to form the observed m *m*/*z* 413.2 is unclear).

**Table 1 molecules-25-05280-t001:** Fragments derived from collision-induced dissociation (CID) on protonated verteporfin (*m*/*z* 719.3). Fragment abundance is defined as MS peak intensity divided by the sum of all fragment peak intensities (Appendix A). The proposed neutral fragment lost from protonated verteporfin is reported for each fragment, together with its chemical structure.

Observed *m*/*z*	Fragment Abundance	Neutral Loss
645.3	63%	methyl acetate	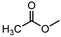
587.3	14%	monomethyl succinate	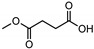
659.3	13%	acetic acid	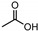
687.3	4.5%	methanol	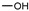
631.3	3.5%	methyl propanoate	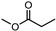
601.3	1.7%	dimethyl oxalate	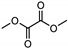

**Table 2 molecules-25-05280-t002:** Observed fragments for protonated verteporfin following gas phase CID and laser photofragmentation. Each photofragment represented by its *m*/*z* (first column) can be observed (✓) or not (**X**) in the CID spectrum (second column), or gas-phase laser excitation at 398 nm (third column) or 758 nm (fourth column). “Low” refers to a photofragment that is present in the mass spectrum but observed at particularly low intensities compared to the other photofragments.

*m*/*z* Fragments	CID	398 nm	758 nm
704.3	**X**	**X**	✓
703.3	**X**	**X**	✓
687.3	✓	✓	✓
659.3	✓	✓	✓
645.3	✓	✓	✓
631.3	✓	✓	✓
613.2	**X**	✓	✓
601.3	✓	✓	✓
587.3	✓	✓	✓
571.2	**X**	✓	✓
557.2	**X**	✓	Low
541.2	**X**	✓	Low
527.2	**X**	✓	Low
513.2	**X**	✓	Low
499.2	**X**	✓	Low
483.2	**X**	✓	Low
469.2	**X**	✓	Low
455.2	**X**	✓	Low
441.2	X	✓	Low

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
