# Peer review of "Photoproducts of the Photodynamic Therapy Agent Verteporfin Identified via Laser Interfaced Mass Spectrometry"

_molecules, 2020, doi:10.3390/molecules25225280_

Round 1

Reviewer 1 Report

The manuscript proposed by Caroline E. H. Dessent et al. (Ref. No.: Molecules 983168) for publication in Molecules, is entitled « Photoproducts of the Photodynamic Therapy Agent Verteporfin Identified via Laser Interfaced Mass Spectrometry ». It concerns Verteporfin (a photosensitizing drug for photodynamic therapy (PDT) with activation by red light at 689 nm) and a direct study of its photo-fragmentation channels in the gas-phase ; this study was “conducted using a laser interfaced mass spectrometer across a broad photo-excitation range from 250-790 nm”. Verteporfin was found to photo-degrade with loss of its carboxyl and ester side-chain functional groups. Solution-phase photolysis products were characterized using electrospray ionization mass spectrometry and assigned with the aid of the gas-phase measurements. It is claimed that the laser-interfaced mass spectrometry is a very useful new tool for characterizing photoproducts and photo-decay pathways, with potential for application in a range of photo-pharmaceutical molecule. The presented results provide new information on the thermal break-down products and photoproducts of this light-triggered drug.

To my point of view, this study is of interest in a potential better understanding of the direct and indirect mechanisms of action of “photo-frug”; it is quite well written, documented and illustrated. The study is rigorous and this manuscript may deserve for publication in Molecules.

Author Response

We thank the reviewer for their supportive comments on the manuscript and note that no changes to the manuscript were required by the reviewer.

Reviewer 2 Report

The paper entitled „Photoproducts of the Photodynamic Therapy Agent Verteporfin Identified via Laser Interfaced Mass Spectrometry” by Chris Furlan, Jacob A. Berenbeim, Caroline E. H. Dessent reports a direct study of Verteporfin photodegradation conducted using a laser interfaced mass spectrometer across a broad photoexcitation range from 250-790 nm. Paper cover missing knowledge of photostability in the porphyrinoid group. The paper is interesting, well written, and can be published after minor revision. Please consider performing photolysis experiments in other solvents, especially in the water. The determination of photodegradation products in the water environment is crucial for assessing the safety of this therapy.

Author Response

We thank the reviewer for their positive review of the manuscript, and for alerting us to the fact that we had omitted to mention previous work on the photostability of porphyrinoids.  To address this we have added the following sentence at line 61, along with the new reference [10].

We note that Berezin and coworkers recently studied the photostability of a group of related porphyrinoids, finding that their photostability depended on the degree of non-planarity of the macrocycle, as well as the symmetry of the protonated form present in proton-donating solvents. [10]

The reviewer makes a reasonable suggestion that it would be worthwhile extending the work reported to further solution-phase photolysis measurements in the water environment, but unfortunately this is not currently possible due to limitations in laboratory access arising due to the COVID pandemic. However, this is certainly something to be investigated in further work.  We chose acetonitrile as the solvent for the photolysis measurements as this was the optimum solvent for electrospraying verteporfin.  We have added the following sentence at line 317 to note this point:

"All solution photolysis experiments were performed in acetonitrile, since this solvent was found to be the optimum solvent for electrospraying verteporfin. "

Reviewer 3 Report

As the title expresses it clearly, this manuscript presents the analysis of the photodissociation of  Verteporfin, used in the treatment of the macular
degeneration via photodynamic therapy.

Fragmentation of Verteporfin is studied extensively by MS, with CID, UV and visible photo fragmentation,  solution photolysis and a comprehensive gaz phase photo depletion study in the whole visible range.

The study is rather complete, the different experimental approaches are compared and the detected fragments are tentatively assigned. Altogether it is a sound work.

I have nevertheless one remark. The therapeutic action of Verteporfin is associated in the litterature to the production of singlet O2 or ROS species. The results do not clearly show any path to O2 production. In the discussion the authors state in line 260 "with the loss of macrocycle conjugation due to reaction with oxygen" for which they have no evidence. In the photolysis experiments they posit the role of molecular Oxygen in the degradation process, but do not test it. They could for instance redo the photolysis experiment in oxygen free conditions and see if they observe any differences.

This should be more discussed.

Additional remarks :

  • I do not understand the word synthesizer in the sentence line 31
  • you cannot say that "BPD-MA is bound to fetal calf solutions" as F C S is not a molecule nor a sur face
  • There is an imprecision on the mass of the verteporfin precursor ion between lines 101 and 107.
  • line 134 use "visible" instead of VIS.
  • figure 3b,it is a pity to cut the spectra at m/z=700 when two important fragments appear at 703 and 704
  • btw please give some hint on what could be those two fragments seen with IR irradiation
  • line 150,  758 nm is IR not visible
  • line 153, comment why the Q_band is said to be 730 -790 nm while the maximum absorption is at 689 nm
  • also, comment why the photo depletion spectra do not extend to the Q-band, even though itis the most relevant region for the therapeutic aspects
  • line 230, the sentence is missleading, is the 413.2 fragment doubly charged?? please detail
  • line 323, the power of the OPO laser should be given in watt.

Author Response

We thank this reviewer for their positive comments and their careful reading of the manuscript.

To address the points raised by the reviewer:

1.  The reviewer states:" the results do not clearly show any path to O2 production. In the discussion the authors state in line 260 "with the loss of macrocycle conjugation due to reaction with oxygen" for which they have no evidence."  Our evidence was the fact stated that photolysis lead to decolorization of the verteporfin solution which must be associated with loss of macrocyle conjugation.  To address this reviewer's comment, we have modified the sentence at line 260 to say:

"ESI-MS analysis of the photolyzed solutions of verteporfin in this study led to the identification of a number of photoproducts, tentatively assigned as products associated with the loss of macrocycle conjugation likely due to reaction with oxygen.  These photoproducts are consistent with the observed discoloration of the solution (photobleaching)."

The reviewer states:

  "In the photolysis experiments they posit the role of molecular Oxygen in the degradation process, but do not test it. They could for instance redo the photolysis experiment in oxygen free conditions and see if they observe any differences."

This is a reasonable suggestion for a further experimental study that explores the solution-phase chemistry in more detail.  However, it was not possible for us to conduct further experiments currently due to limitations in laboratory access due to the COVID pandemic.  To note this, we have inserted a sentence at line 262 in the manuscript:

"(Future experiments performed on solutions in the absence of oxygen will be useful to test this hypothesis.)"

Further points raised by the reviewer:

i. "I do not understand the word synthesizer in the sentence line 31" We have replaced synthesizer by producer

ii. "you cannot say that "BPD-MA is bound to fetal calf solutions" as F C S is not a molecule nor a surface" We have modified the sentence (line 47) to clarify that the BPD-MA binds to proteins in the fetal calf solutions.

iii. "There is an imprecision on the mass of the verteporfin precursor ion between lines 101 and 107." We have corrected the mass at line original 107 (table heading) which was a typo.

iv. "Line 134 use "visible" instead of VIS."  This has been done (now at line 137).

v. "Figure 3b,it is a pity to cut the spectra at m/z=700 when two important fragments appear at 703 and 704."  We have remade this figure so that an expanded mass range is visible.

vi. "btw please give some hint on what could be those two fragments seen with IR irradiation" We assume that the reviewer is referring to the fragments seen upon 689 nm irradiation. (In the accepted standard, this wavelength is generally considered to be a visible wavelength.) We refer the reviewer to Figure 6 where we have given tentative structures.

vii. "line 150,  758 nm is IR not visible" We have modified this sentence to acknowledge that this wavelength is on the visible-IR borderline: The sentence (now line 153) now reads...visible/IR photon excitation 

viii. "line 153, comment why the Q_band is said to be 730 -790 nm while the maximum absorption is at 689 nm." We assume the reviewer is referring here to the maximum in the SI solution-phase spectrum. We refer the reviewer here to the discussion at current lines 171-174, where differences between the solution and gas-phase band maxima is discussed. 

ix. "Also, comment why the photo depletion spectra do not extend to the Q-band, even though it is the most relevant region for the therapeutic aspects"  This work was not performed as part of this study, but the reviewer is correct in suggesting that it would be desireable in a future study.  We note that the scan range we have covered does incorporate the second most intense feature of the Q band (see Figure S2).

x. "line 230, the sentence is missleading, is the 413.2 fragment doubly charged?? please detail".  The Figure 6c gives a tentative structure for a molecule with m/z 412.2.  In the experiment, we detect a fragment with m/z 413.2, hence it must correspond to the drawn m/z 412.2 structure with one additional H atom, as the figure caption describes.  (We are not sure why the reviewer mentions doubly charged fragments.)  It was unclear where this additional H would attach to the drawn fragment so we have presented it as shown. At line 213, we have added the following clarification:

"(The lowest energy position for this H atom to attach to form the observed m/z 413.2 is unclear.)"

 xi. "line 323, the power of the OPO laser should be given in watt."  As the OPO is a pulsed laser, we have quoted the energy in mJ as this is standard practice. In considering this comment, we noticed that we had not given the laser details in the methods section.  We have therefore added:

"Photons were produced by an Nd:YAG (10 Hz, Surelite, Amplitude Laser Group, San Jose, CA, USA) pumped OPO (Horizon, Amplitude Laser Group, San Jose, CA, USA) laser."